# Undifferentiated Carcinoma with Osteoclast-Like Giant Cells of the Common Bile Duct: A Case Report of a Rare Entity at an Unusual Location

**DOI:** 10.3390/diagnostics12071517

**Published:** 2022-06-21

**Authors:** Chuan-Han Chen, Hsin-Ni Li

**Affiliations:** 1Department of Radiology, Taichung Veterans General Hospital, Taichung 40705, Taiwan; bonanza0622@gmail.com; 2Department of Pathology and Laboratory Medicine, Taichung Veterans General Hospital, Taichung 40705, Taiwan

**Keywords:** UCOGC, EHBDs, CBD, CHD, pancreas

## Abstract

Undifferentiated carcinoma with osteoclast-like giant cells (UCOGC) is a rare variant of carcinoma with unique radiological and pathological features. This unusual carcinoma has been reported in a variety of organs and pancreas is the most frequently involved anatomical site. UCOGC of pancreas attains a relatively indolent clinical behavior and should be distinguished from ordinary pancreatobiliary adenocarcinoma. This paper presents the first case of UCOGC involving the entire segment of common bile duct (CBD) and common hepatic duct (CHD) without extending to the pancreatic tissue. Getting familiar with its clinical, radiological and pathological characters can help establish accurate diagnosis despite the occurrence of an unusual location.

## 1. Introduction

Carcinoma of the extrahepatic bile ducts (EHBDs) is rare, varies from 0.53 to 2 cases per 100,000 person years worldwide, with higher frequency in Asian countries [1]. Chronic inflammatory conditions such as sclerosing cholangitis, choledocholithiasis and choledochal cyst are known predisposing factors. In the fifth edition of World Health Organization (WHO) Classification of Digestive System Tumours, EHBDs are further categorized into four subtypes, namely bile duct carcinoma (cholangiocarcinoma), adenosqaumous carcinoma, squamous cell carcinoma and undifferentiated carcinoma. The prognosis of EHBDs largely depends on the stage of disease presentation and resectability is another key factor associated with outcome.

Tumors with osteoclast-like giant cells have been reported within variable primary sites including the skin, breast, thyroid gland, lung, uterus, periampullary region and pancreas [2,3,4,5,6,7,8,9,10,11]. With scarce case reports and case series, the association between the presence of the osteoclast-like giant cells and tumor prognosis remains elusive. Previous impression of carcinomas with osteoclast-like giant cells is that they are highly aggressive tumors composed of pleomorphic neoplastic cells [12,13,14]. However, recent studies reveal that undifferentiated carcinoma with osteoclast-like giant cells (UCOGC) of the pancreas possesses significant better prognosis than that of conventional ductal adenocarcinoma and has been distinguished from undifferentiated carcinomas of the pancreas in the WHO classification. Herein, we report a case of UCOGC in the common bile duct (CBD) and common hepatic duct (CHD). To our knowledge, this is the first case report of UCOGC involving the entire segment of CBD and CHD without pancreatic duct extension. Accurate diagnosis is important as it may suggest more indolent clinical behavior as well help guide the subsequent appropriate management.

## 2. Case Description

A 66-year-old female had a medical history of recurrent intrahepatic duct (IHD) stones underwent choledocholithotomy, hepaticolithotomy, cholecystectomy, left lateral sectionectomy of the liver due to left IHD orifice stenosis and left hepaticojejunostomy 17 years ago. 

She suffered from abdominal pain in the right upper quadrant along with right flank and back discomfort for 2–3 months. The symptoms exacerbated and she went to a local hospital where abdominal computed tomography (CT) revealed dilatation over IHD and CBD in the presence of stones. Acute cholangitis was impressed and antibiotics were prescribed. The symptoms of fatigue, poor appetite, jaundice, low-grade fever and right epigastric pain developed again in spite of therapy. She visited our Emergency Department for medical help, where cholangitis was suspected. She was subsequently admitted for further evaluation and management. Endoscopic retrograde cholangiopancreatography (ERCP) was performed which showed hemobilia in the presence of lots of tissue material and blood clots within CBD. Some tissue material was extracted and sent to pathologic examination. Magnetic resonance cholangiopancreatography (MRCP) was arranged, which revealed dilatation and mild increase in wall thickness and enhancement of the biliary tract. A lobulated lesion in CHD and CBD was also found, extending to the confluence (about 9.8 cm × 1.9 cm), with heterogeneous signal intensities on the T1-weighted image (mainly high signal intensity) and T2-weighted image (mainly low signal intensity). Some equivocal enhancing foci of the lesion were suspected (Figure 1). The impression of cholangitis could be compatible, accompanied with possible debris, hematoma or neoplasm. The pathology reported an admixture of blood clots, fibrin and inflammatory cells. Aggregation of histiocytes and scanty pancreatobiliary-type epithelium with bland cytologic features was noted whilst there was no evidence of malignancy. A blood culture yielded Escherichia coli. Empirical antibiotics were prescribed and the symptoms subsided. The patient was discharged under a stable condition. 

However, her symptoms recurred four months later. She visited our Emergency Department again where the abdominal CT showed an intramural lesion in CHD and CBD with mixed high and low densities and heterogeneous enhancement, accompanied with dilatation of the biliary tract (Figure 2). The picture was similar to that on the prior MRCP. She was treated as cholangitis and ERCP biopsy was redone for the neoplastic survey. During the procedure, CBD dilation with tissue retention and papillary mucosal surface were noted. Similar to the prior pathological report, descriptive diagnosis illustrating blood clots, histiocytes and some multinucleated giant cells aggregate was made. Antibiotics were used to treat the patient effectively. 

Unfortunately, the patient experienced right epigastric pain, jaundice and low-grade fever 10 months after her first presentation. Elevated levels of alkaline phosphatase (403 U/L), total bilirubin (2.1 mg/dL), direct bilirubin (1.36 mg/dL) and CA 19-9 (10.94 Unit/mL) were noted. Aspartate aminotransferase (38 U/L), alanine aminotransferase (43 U/L), CEA (2.05 ng/mL) and AFP (2.19 ng/mL) were within normal ranges. The abdominal CT and MRCP were undertaken which showed similar findings as prior images, but a progression in biliary tract dilation and an enlarged lesion size (about 11.3 cm × 3.2 cm), with more conspicuous cystic parts and heterogeneous enhancement (Figure 3). Neither enlargement of the regional lymph nodes nor invasion to the adjacent tissue were found.

Under the impression of recurrent cholangitis coexisted with an uncertain CBD neoplasm, the patient underwent CBD tissue removal by choledochoscope. 

Grossly, a bloody mass composed of a mixture of tissue debris, blood clots and pus was extracted from the bile ducts. Pathology revealed a tumor consisting of three cell types including osteoclast-like multinucleated giant cells (OGCs), mononuclear histiocytes and neoplastic mononuclear cells in a background of extensive hemorrhage and deposition of hemosiderin pigments (Figure 4A–D). Foci of tissue necrosis and inflammatory exudates were noticed (Figure 4E). A microscopic focus of malignant epithelia exhibiting papillary and tubular glandular structure floating in the blood clots was found (Figure 4C). There was no microscopic evidence of tumor invasion to the stroma of the bile duct.

Immunohistochemistry (IHC) was performed on selected blocks and showed that malignant epithelia were highlighted by AE1/AE3 (Figure 4F) and CK19 (Figure 4G). Weak positivity was also observed at some atypical mononuclear neoplastic cells. The osteoclast-like giant cells and histiocytic mononuclear cells were positive for CD68 (Figure 4H). P53 was expressed in few neoplastic cells (Figure 4I).

After the surgery, the patient’s condition was improved. The level of alkaline phosphatase (142 U/L), total bilirubin (0.55 mg/dL) and direct bilirubin (0.19 mg/dL) returned to the normal limits. The patient did not receive adjuvant chemotherapy and exhibited no evidence of recurrence after one year of follow-up.

## 3. Discussion

Undifferentiated carcinomas with osteoclast-like giant cells (UCOGC) have been described in various organs with limited case numbers and reviews. Extrahepatic biliary tree occurrence of this tumor is even rare. As a result, WHO did not include UCOGC as a distinct disease entity in the extrahepatic biliary system. Despite the rarity, few case reports and case series have described with similar medical conditions which were summarized in Table 1. For example, a case report revealed that a polypoid UCOGC arose and grossly confined in the lumen of the distal CBD [15]. In addition, a case series with four extrahepatic biliary tree giant cell tumors collected showed that two patients had tumors located at distal common bile duct, one in the cystic duct (CD) and one in the gallbladder. The patients with CD or CBD tumors presented with biliary obstruction [16]. All these reported cases of UCOGC in extrahepatic bile duct had tumor at the distal or middle CBD. We herein report the first case of UCOGC of extrahepatic bile duct involving the entire segment of CBD and CHD, highlighting the diverse anatomical distribution of this unusual entity. The clinical presentation (jaundice and fever) of this case was further complicated with the concurrent bacterial cholangitis, leading to the missed diagnosis initially.

The imaging features of UCOGC have not been well described due to the rarity of this entity. Few case reports and series have been reported mainly focusing on pancreatic UCOGC, in which a larger tumor size containing a cystic component was noticed compared with conventional pancreatic ductal adenocarcinoma [20]. Additionally, dilatation of the bile duct and pancreatic duct along with the presence of necrotic areas were commonly observed [21,22]. Occasional calcification, hemorrhage and venous tumor thrombus had also been recorded [23,24]. On magnetic resonance imaging (MRI), UCOGC usually presented a well-defined hypovascular lesion with low signal intensity on T1- and T2-weighted images, probably resulting from hemosiderin deposits, and relatively high signal intensity in the central area, probably reflecting the necrotic part of it [25]. Despite its large volume and rapid tumor growth, tissue infiltration and lymph node metastases were not common, with relatively low malignancy and late clinical symptoms [26]. In a case report of UCOGC of the distal CBD, a polypoid well circumscribed, cystic, rapid growing mass with hemorrhage in the ductal lumen was depicted [15]. In keeping with their findings, our case showed a huge (9.8 cm × 1.9 cm) lobulated, relatively well-defined hemorrhagic mass in the dilated CHD and CBD, with heterogeneous enhancement. On the follow-up images, it revealed rapid growth (11.3 cm × 3.2 cm 9 month later) with more prominent cystic parts and dilatation of the biliary tract. Neither definite evidence of invasion to the adjacent tissue nor enlargement of the regional lymph nodes were discovered, which was consistent with the features of this entity described in prior literatures.

Pathologically, the characteristic picture of UCOGC has been well documented [16,20,25]. In most cases, the compact and cellular neoplasm resembled giant cell tumors of bone in a background of hemorrhage and deposition of numerous coarse hemosiderin pigments. The tumor was composed of three distinct cell populations: OGCs, mononuclear histiocytes and mononuclear carcinoma cells. The OGCs displayed the characteristic features of this cell type seen in the bone, including large cells with acidophilic cytoplasm and multiple uniform nuclei, usually clustering in the central aspect of the cell. The OGCs showed positive immunoreactivity for CD68 but a lack of reactivity for epithelial markers. They were commonly found in areas adjacent to the hemorrhage and necrosis. The mononuclear histocytes, as the name implies, had monotonous single nuclei and were immunoreactive for histiocytic marker CD68. Epithelial markers such as cytokeratin were generally negative in mononuclear histiocytes. The mononuclear neoplastic cells, on the contrary, were positive for cytokeratin, confirming biliary epithelial origin. The neoplastic cells were spindly to squamoid in appearance and could demonstrate varying degrees of cytologic atypia and mitotic activity [16,20,25]. As studies presumed that tumors containing osteoclast-like giant cells spanned the spectrum from truly benign giant cell tumors to highly lethal anaplastic spindle and giant cell carcinomas, careful histologic and cytologic examination and IHC studies were recommended [16,27].

Conventional carcinoma or precursor component could be found, providing evidence of the relationship of UCOGC to the underlying carcinoma. In the pancreas, UCOGC was identified in association with intraductal papillary mucinous neoplasm (IPMN) [20] and mucinous cystic neoplasm (MCN) [28]. It is noteworthy that the carcinomatous part was not always present, particularly in small biopsies or aspiration, making correct diagnosis difficult initially. Thus, the presence of osteoclast-like giant cells and mononuclear cells in limited samples should raise the possibility of UCOGC. In addition, extensive sampling as well as the use of epithelial and histiocytic markers are recommended to avoid diagnostic pitfalls [29]. The importance of extensive sampling was also emphasized by Luchini et al., who discovered that the most important criterion for prognosis was the presence of an associated epithelial neoplasms [27].

Despite the similar or identical histological pictures, the role that infiltrated osteoclast-like giant cells plays in terms of prognosis remains controversial. These cells were believed to be non-neoplastic and were attracted via mechanisms yet to be discovered. Osteoclast-like giant cells found in extraskeletal tumors appeared to be a specific type of macrophages which was distinct from both osteoclasts and foreign-body giant cells [30]. In the breast, some suggested that the presence of osteoclast-like giant cells indicates a less aggressive tumor with a better outcome [31], while some had the opposite notion [32]. In the pancreas, previous literature regarded that UCOGC was a highly aggressive tumor with an even worse prognosis than ordinary pancreatic ductal adenocarcinoma [12,13,14]. The concept changed when Muraki et al. analyzed the clinicopathologic features in 38 cases of UCOGC and concluded that the patients had a more protracted clinical course than previously believed [20]. The better prognosis of patients with UCOGC might be attributable to its intraductal growth and the unique morphology which represented immune response to an underlying carcinoma [20,27].

More than 20 years ago, Westra et al. demonstrated that KRAS oncogene mutation, which was highly characteristic of pancreatic adenocarcinoma [33,34], was identified in a majority of UCOGC of pancreas (seven out of eight), suggesting a common route to malignant transformation [35]. Using whole exome sequencing of eight UCOGC of the pancreas, a recent study demonstrated that genetic alterations observed in UCOGC were similar to those identified in carcinogenesis of pancreatic ductal adenocarcinoma, including activating the oncogene KRAS and inactivating the tumor suppressor genes CDKN2A, TP53 and SMAD4 [25,27,36]. The two abovementioned publications supported a shared pathway of tumorigenesis in ordinary pancreatic carcinoma and UCOGC. As genetic alterations vary widely among different malignancies, it is still unclear whether UCOGC of various anatomical sites have a similar phenomenon. Interestingly, our case had medical history of recurrent IHD stones status post-surgery and recurrent cholangitis, which were risk factors for carcinoma of EHBDs. The patient developed UCOGC of CBD and CHD afterwards, which supported the assumption that UCOGC of this site had the same predisposing factors as typical carcinoma of this anatomical location. Two prior case reports also indicated the association between giant cell tumor-like cholangiocarcinoma and cholelithiasis/hepatolithiasis [37,38]. In addition, next-generation sequencing detected mutations of KRAS and PIK3CA in a case of UCOGC arising in distal CBD, keeping in line with similarity to conventional cholangiocarcinoma [15]. Additional case studies are needed to help better define the biology of these neoplasms.

Due to the rarity of UCOGC, treatment options have never been standardized. In the UCOGC of the pancreas, surgery is the first-choice treatment. The efficacity of radio-/ and/or chemotherapy remains to be evaluated [39]. Recently, PD-1 and PD-L1 immunotherapy has emerged as a promising therapeutic choice in several malignancies. As PD-L1 is frequently expressed in the UCOGC of the pancreas, particularly in cases with an associated ductal adenocarcinoma [40,41], PD-L1 immunotherapy may provide a potential therapeutic strategy to treat UCOGC in the future.

## 4. Conclusions

We have reported a rare carcinoma in an unusual anatomical site, the UCOGC of the entire CBD and CHD. This unique case betters our understanding of the heterogeneity of pancreatobiliary carcinomas. Getting familiar with its clinical, radiologic and pathologic characters directs an early and accurate diagnosis and helps guide the subsequent appropriate management.

## Figures and Tables

**Figure 1 diagnostics-12-01517-f001:**
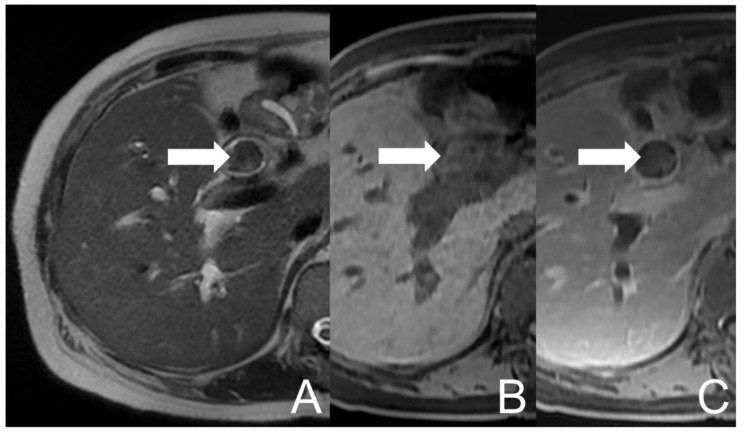
UCOGC of CHD and CBD in a 66-year-old female. The initial axial MRI images show a mass (arrow) with heterogeneous signal intensities in CBD, demonstrating relatively low signal intensity on the T2-weighted image (**A**) and relatively high signal intensity on the pre-contrast T1-weighted fat-saturated image (**B**), with faint enhancing foci on the post-contrast T1-weighted fat-saturated image (**C**). The imaging findings are suggestive of a neogrowth with hemorrhage.

**Figure 2 diagnostics-12-01517-f002:**
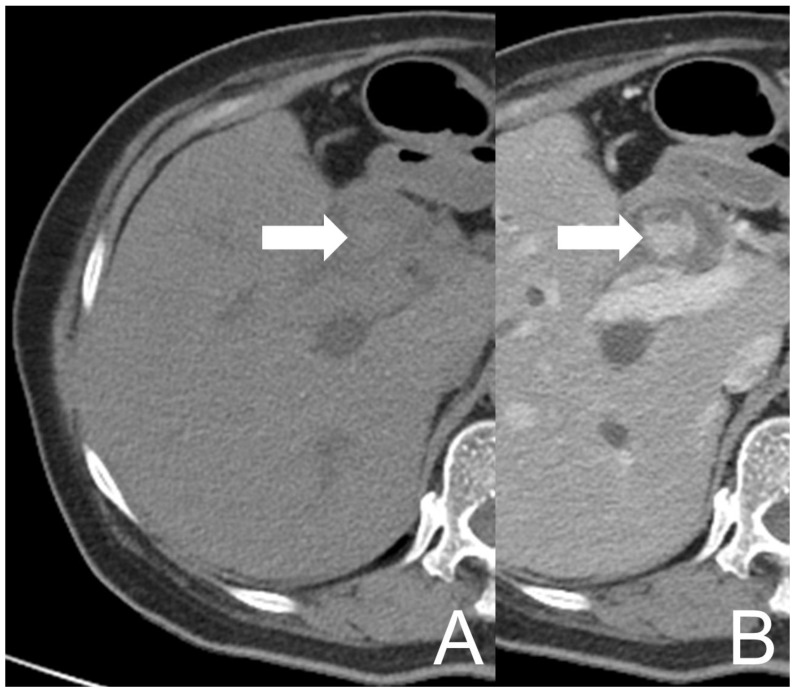
UCOGC of CHD and CBD in a 66-year-old female, 4 months after the initial presentation. The axial CT images show a mass (arrow) in CBD, demonstrating relatively high density on the pre-contrast image (**A**) and heterogeneous enhancement on the post-contrast image (**B**). The imaging findings are compatible with a neogrowth with hemorrhage.

**Figure 3 diagnostics-12-01517-f003:**
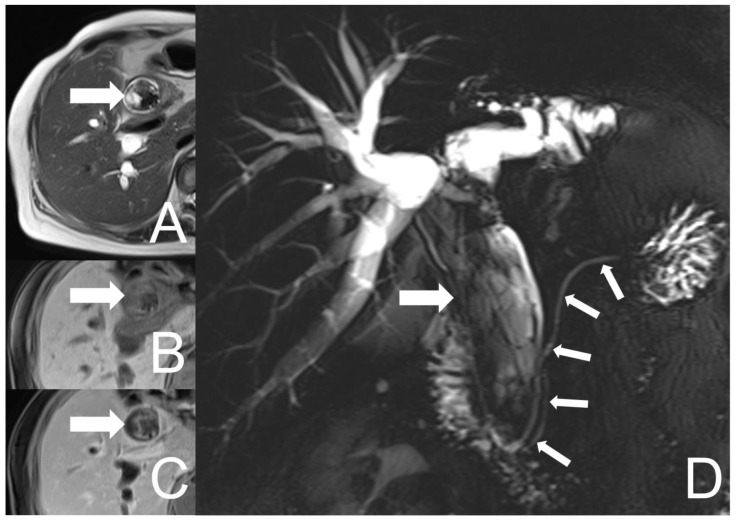
UCOGC of CHD and CBD in a 66-year-old female, 10 months after the initial presentation. The tumor (thick arrow) shows an increase in size with more conspicuous cystic parts on the axial T2-weighted image (**A**), and more conspicuous heterogeneous signal intensities and enhancement on the axial pre- and post-contrast T1-weighted fat-saturated images (**B**,**C**). The hypointensities of the lesion on both T1- and T2-weighted images could be suggestive of hemosiderin deposition. The MRCP (**D**) reveals the tumor occupying the whole CHD and CBD, with the dilatation of the biliary tract. Neither tumor involvement nor dilatation of the pancreatic duct were found (thin arrows).

**Figure 4 diagnostics-12-01517-f004:**
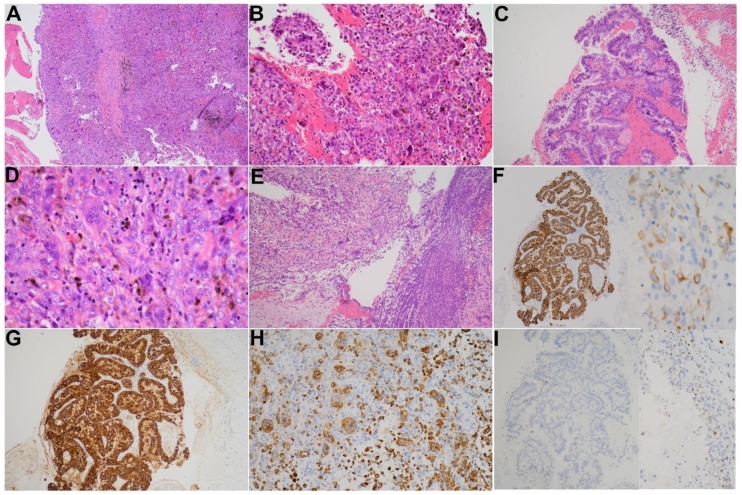
Pathological features of this case. (**A**) Hypercellular tumor in a background of hemorrhage and numerous hemosiderin pigments (H&E stain, ×40). (**B**,**D**) The tumor is composed of three cell types including OGCs, mononuclear histiocytes and neoplastic mononuclear cells (H&E stain, ×100 and ×400, respectively). (**C**) Malignant epithelia exhibiting papillary and tubular glandular structure (H&E stain, ×100). (**E**) Foci of tissue necrosis and inflammatory exudates (H&E stain, ×100). (**F**) Immunohistochemistry of AE1/AE3 highlights the malignant epithelia and mononuclear neoplastic cells (×100). (**G**) Immunohistochemistry of CK19 highlights the malignant epithelia (×100). (**H**) Immunohistochemistry of CD68 highlights the osteoclast-like giant cells and mononuclear histiocytes (×100). (**I**) Immunohistochemistry of p53 is expressed in some neoplastic cells (×100).

**Table 1 diagnostics-12-01517-t001:** Clinical features of cases with UCOGC of CBD.

Case No.	Age/Sex	Clinical Presentation	Location	Gross Features	Treatment	Follow-Up	Reference
1	56/M	Jaundice, abdominal pain	Distal CBD	1.6 cm, polypoid	Whipple procedure	Free of disease 50 months after surgery	[16]
2	60/M	Obstructive jaundice	Distal CBD	1 cm, polypoid	Whipple procedure	No	[16]
3	60/M	Asymptomatic	Distal CBD	1.5 cm, polypoid	Whipple procedure	Free of disease 16 months after surgery	[17]
4	73/F	Abdominal pain	Distal CBD	2.5 cm × 0.7 cm	Pylorus-preserving pancreas head resection and pancreaticogastrostomy	No	[18]
5	81/M	Jaundice	Middle CBD	1.5 cm × 0.6 cm, polypoid	Excision of the gallbladder and extrahepatic bile duct and a Roux-en-Y cholangiojejunostomy	Free of disease 6 months after surgery	[19]
6	66/F	Jaundice and fever	Entire CBD and CHD	11.3 cm, lobulated	Choledocoscopic tissue removal	Free of disease 12 months after surgery	Our case

## Data Availability

Not applicable.

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
