# Peer review of "Undifferentiated Carcinoma with Osteoclast-Like Giant Cells of the Common Bile Duct: A Case Report of a Rare Entity at an Unusual Location"

_diagnostics, 2022, doi:10.3390/diagnostics12071517_

Round 1
Reviewer 1 Report
This is a nicely written case report of an interesting case and a short literature review on the topic.
A minor comment: please consider adding some references in the second paragraph of introduction, where you describe the prognosis of UCOGC in the pancreas.
Also, in the Discussion, in the part you elaborate more on the same topic, you could mention that in the past it was believed that pancreatic UCOGC had a poor prognosis (with a couple of references), just before you mention about the recent studies showing UCOGC behaves actually better.
Thanks a lot for your contribution!
Author Response
Response to Reviewer 1 Comments:
This is a nicely written case report of an interesting case and a short literature review on the topic.
Response: Many thanks for the compliments.
A minor comment: please consider adding some references in the second paragraph of introduction, where you describe the prognosis of UCOGC in the pancreas.
Response 1: Many thanks for the useful suggestions. We have added a couple of references in the second paragraph of introduction where we addressed the prognosis of UCOGC in the pancreas.
Also, in the Discussion, in the part you elaborate more on the same topic, you could mention that in the past it was believed that pancreatic UCOGC had a poor prognosis (with a couple of references), just before you mention about the recent studies showing UCOGC behaves actually better.
Thanks a lot for your contribution!
Response 2: Many thanks for your useful suggestions. In the Discussion section (the 5th paragraph), we have briefly mentioned the dismal prognosis of pancreatic UCOGC which was previously described in the literature before recent studies. The revision makes the article more comprehensible and we do appreciate your advice.
Reviewer 2 Report
This papar desribes a very rare case of the bile duct tumor, Undifferentiated carcinoma with osteoclast-like giant cells. The images of the tumor and histopathology are interesting and this paper is worth reading for clinicians.
One thing I ask the author is to provide a picture of non-enhanced T1 WIs of MRI. Because a picuture of non-enhanced T1 WIs is better to give us information about the histopathological background of the tumor than that of enhanced T1 WIs.
Author Response
Response to Reviewer 2 Comments:
This papar desribes a very rare case of the bile duct tumor, Undifferentiated carcinoma with osteoclast-like giant cells. The images of the tumor and histopathology are interesting and this paper is worth reading for clinicians.
Response: Many thanks for the compliments.
One thing I ask the author is to provide a picture of non-enhanced T1 WIs of MRI. Because a picuture of non-enhanced T1 WIs is better to give us information about the histopathological background of the tumor than that of enhanced T1 WIs.
Response 1: Many thanks for your suggestions. We have amended Figure 3 by adding non-enhanced T1 WIs of MRI (Figure 3B) and inserting the figure legends accordingly. The non-enhanced T1 WIs, as you mentioned, suggested hemorrhage and deposition of hemosiderin pigments as what we saw in the histological pictures.